# Ex Vivo Propagation of Pinctada Birnavirus Using Mantle Tissue Fragment Culture: Application for Measuring Replication at Different Temperatures, TCID_50_ Assay, and UV Sensitivity

**DOI:** 10.3390/pathogens14010076

**Published:** 2025-01-15

**Authors:** Tomomasa Matsuyama, Takashi Atsumi, Ikunari Kiryu, Kousuke Umeda, Natsuki Morimoto

**Affiliations:** 1Japan Fisheries Research and Education Agency, Pathology Division, Aquaculture Research Department, Fisheries Technology Institute, Minami-Ise 516-0193, Mie, Japan; 2Mie Prefecture Fisheries Research Institute, Shima 517-0404, Mie, Japan

**Keywords:** Pinctada birnavirus, PiBV, *Pinctada fucata*, mass mortality, summer atrophy, temperature, TCID_50_, UV, tissue culture

## Abstract

Pinctada birnavirus (PiBV) is the causative agent of summer atrophy in pearl oyster (*Pinctada fucata* (Gould)). The disease, which induces mass mortality in juveniles less than 1 year old and abnormalities in adults, was first reported in Japan in 2019. Research on the disease has been hindered by the lack of cell lines capable of propagating PiBV. We established an ex vivo method for PiBV propagation using mantle tissue, the primary infection site of the virus. The method was used to investigate the proliferation characteristics of the virus at different culture temperatures and the sensitivity of the virus to UV radiation. The marginal zone of the mantle was found to be the most suitable for PiBV replication in terms of both viral yield and reproducibility. PiBV showed optimal propagation at an incubation temperature of 25 °C, with minimal to no increase at 15 °C or 32.5 °C. Using the tissue culture infectious dose 50 (TCID_50_) measurement system developed in this study, we found that PiBV propagation was no longer detectable after UV irradiation at 6150 J/m^2^ or higher. The tissue fragment culture method developed in this study is expected to facilitate both ex vivo experiments and PiBV research.

## 1. Introduction

Pinctada birnavirus (PiBV) is the tentative name for an unclassified virus that was identified as the causative agent of a mass mortality event referred to as ’summer atrophy’ in juvenile pearl oysters (*Pinctada fucata* (Gould)) in Japan [1,2]. Phylogenetic analysis suggests a close relationship between PiBV and the genus *Entomobirnavirus*, which primarily infects insects [2]. In affected individuals, the soft body tissues atrophy toward the dorsal margin. While most infected adults recover from the atrophy, surviving individuals show symptoms of what is referred to as shell disease [1,3], which is characterized by melanin deposition in the nacreous layer of the shell [4]. The virus infects the outer epithelial cells of the mantle [2]. In contrast to members of the genus *Aquabirnavirus*, which predominantly infects fish and has been isolated in bivalve mollusks, PiBV could not be propagated in fish cell lines (EPC-F, EK-1, SSN-E11, BF-2, CHSE-214, RTG-2). Attempts to propagate PiBV in insect cell lines (Sf9, High Five, and Drosophila S2) have also been unsuccessful. The inability to use cell lines for viral culture presents a significant limiting factor in the study of viruses that infect mollusks, including PiBV.

Various cell lines have been established from vertebrates and arthropods for use as alternative models to live organisms for research purposes. Since the Nobel Prize-winning poliovirus research by Enders, Robbins, and Weller in 1949 [5], cultured cell lines have become widely used in virology. These cell lines have facilitated the isolation and propagation of viruses and enabled in vitro research. Additionally, large-scale virus production using cultured cells has become essential for vaccine development. Cultured cell lines are easy to handle and allow for highly reproducible experimental results under controlled laboratory conditions, making them indispensable tools in virology.

However, with the exception of some arthropods, cell lines have not been established for most invertebrates [6]. In mollusks, which include many species that are economically important, numerous studies have attempted to establish cultured cell lines [6,7,8]. However, the only cell line that has been successfully established is an embryonic cell line (Bge) from the air-breathing freshwater snail *Biomphalaria glabrata* [9]. Consequently, very few studies on viruses that infect mollusks have been conducted using cultured cell lines. In some cases, primary cells harvested directly from mollusks are used as an alternative for virus isolation [10,11,12,13,14,15]. Primary cultured cells retain the characteristics of in vivo cells, which immortalized cell lines often lose during the immortalization process [16]. However, primary cells must be prepared from live organisms for each experiment. Except for hemocytes, isolating specific cells from tissues requires labor-intensive procedures, such as prolonged enzymatic treatment [17,18]. While hemocytes can be easily isolated in a dispersed state, the virus under study does not always infect them [10,13]. Moreover, due to their lack of division and short survival time in vitro [15], hemocytes are not always suitable for virus research.

On the other hand, in contrast to primary cultured cells, tissue fragments are easier to harvest from living organisms and are less likely to undergo the functional changes often observed in immortalized cultured cell lines. Ex vivo cultivation methods using tissue cultures have been previously reported, such as in studies of Ostreid herpesvirus-1 (OsHV-1) [15]. The mantle, which PiBV infects, has a large surface area and can be readily cut into small tissue fragments. If PiBV can be propagated using mantle tissue fragment culture, it could provide an ex vivo model for virus research as an alternative to cultured cells. In this study, to optimize the conditions for ex vivo cultivation, we evaluated PiBV propagation in three histologically distinct regions of the mantle, i.e., the marginal, pallial, and central zones, and assessed the effect of the rearing water temperature on PiBV propagation using mantle fragment cultures. Furthermore, we investigated the optimal growth temperature for PiBV, established a method for measuring the tissue culture infectious dose 50 (TCID_50_), and examined the conditions for viral inactivation by UV irradiation using this ex vivo propagation method.

## 2. Materials and Methods

### 2.1. Pearl Oysters

Three-year-old pearl oysters, reared in a summer atrophy-free area, were used for the study. Their shell widths ranged from 62 to 83 mm, with an average and standard error of 70.9 ± 3.0 mm (N = 8). The oysters were reared in 65 L tanks containing 56 L of running sea water (approximately 250 mL/min) filtered through a 1 μm pore size filter and maintained at 16–18 °C or 23–25 °C under a natural photoperiod at the laboratory of the Fisheries Technology Institute, Minami-Ise, Mie Prefecture, Japan. As food, 500 mL of cultured diatoms (*Chaetoceros neogracile*, approximately 5 × 10^5^ cells/mL) were added to each tank five times a week, both during the acclimation period and the infection tests.

### 2.2. Reverse Transcription Quantitative PCR (RT-qPCR)

The quantification of PiBV was performed based on the method targeting segment A of the viral genome, as described in a previous report [2]. Briefly, total RNA was extracted from the samples using TRIzol LS Regent (Invitrogen, Waltham, MA, USA) and subsequently dissolved in nuclease-free water. Next, the PiBV genome was quantified using the Thunderbird Probe One-Step qRT-PCR Kit (Toyobo, Osaka, Japan). The reaction mixture was prepared according to the manufacturer’s instructions, using 2 μL of total RNA as a template, and the PiBV genomic RNA was quantified using a LightCycler 96 (Roche, Basel, Switzerland) with the following RT-qPCR cycling conditions: 10 min at 50 °C and 60 s at 95 °C, followed by 40 cycles of 15 s at 95 °C and 45 s at 60 °C. A standard curve was prepared using a pCR2.1 TOPO vector (Thermo Fisher Scientific, Waltham, MA, USA) containing a fragment of segment A. A threshold cycle (Ct) cutoff of less than 36 was applied. The concentration of PiBV genomic RNA in each culture supernatant was calculated using the following formula and expressed as the number of PiBV copies per μL of culture seawater: (Number of virus copies detected × Total amount of RNA extracted from the sample)/Amount of RNA used in the reaction mixture.

### 2.3. Infection Source

The infection source was prepared by re-propagating PiBV ex vivo in mantle tissue culture from a viral stock refrigerated during a previous study. [2]. Briefly, the entire mantles of 10 healthy pearl oysters were collected and placed in a plastic container containing 160 mL of seawater with 1000 units/mL of penicillin G and 1000 μg/mL of streptomycin sulfate (hereafter referred to as culture seawater). A 500 μL aliquot of the infection source solution, which was prepared in a previous study [2] and stored under sterile conditions at 4 °C for 207 days, was then added to the container, and the mixture was cultured at 25 °C for 8 days. The lid of the container was opened slightly to permit ventilation. After the culture period, the mantle tissue was removed, and the remaining solution was filtered through a 100 μm nylon mesh. Vertrel XF (Mitsui Chemicals, Tokyo, Japan) was then added to the filtrate at one-fifth the volume, and the mixture was stirred at 500 rpm for 5 min. The mixture was then centrifuged at 8400× *g* for 10 min, and the supernatant was collected. These steps, i.e., Vertrel XF treatment and centrifugation, were repeated another two times. The final supernatant was layered onto a 20–60% CsCl gradient in TN buffer (10 mM Tris-HCl, 0.8% NaCl, pH 7.8) and centrifuged at 150,000× *g* for 3 h using an SW32Ti rotor (Beckman Coulter, Brea, CA, USA). The CsCl interface was collected, dialyzed with autoclaved seawater using dialysis tubes (MWCO 6–8 kDa; Repligen, Waltham, MA, USA), and penicillin G and streptomycin sulfate were added at final concentrations of 1000 units/mL and 1000 μg/mL, respectively, and then filtered through 0.2 μm filters (Thermo Fisher Scientific, Waltham, MA, USA). The resulting solution was adjusted to 1.0 × 10^4^ copies/μL with culture seawater and used as the infection source. The infection source was stored at 4 °C until use.

### 2.4. Mantle Regions Suitable for PiBV Ex Vivo Propagation

To determine which regions of the mantle are suitable for PiBV ex vivo propagation, the mantles were collected from the pearl oysters (N = 4) that had been reared at 23–25 °C. After washing the mantles with sterile seawater, three tissue fragments measuring approximately 3 mm wide and 5 mm long were excised from the marginal, pallial, and central zones, as shown in Figure 1. Each tissue fragment was placed into a separate well of a 24-well plate. To each well, 1 mL of culture seawater and 10 μL of the infection source were added, and the plate was then incubated at 25 °C under static conditions. RNA was extracted from 50 μL of the culture medium using Trizol LS on days 2, 4, 6, and 9 of cultivation, and the amount of PiBV genomic RNA was measured using RT-qPCR.

### 2.5. Reproducibility of PiBV Propagation Using the Marginal Zone of the Mantle

To assess the reproducibility of PiBV propagation, the entire marginal zone from the left and right mantles of each of the four pearl oysters was divided into 12 sections and tested for ex vivo PiBV infection. Oysters that had been reared at 23–25 °C were used. Each fragment was placed into a separate well of a 24-well plate. To each well, 1 mL of culture seawater and 10 μL of the infection source were added, and the plates were incubated at 25 °C for 7 days. The PiBV genomic RNA in the cultured seawater was then quantified by RT-qPCR.

### 2.6. Effects of Rearing Water Temperature and Incubation Temperature on PiBV Proliferation

To examine the effect of the oyster-rearing water temperature on PiBV propagation ex vivo, tissue samples were collected from oysters reared at two different temperature ranges. In addition, to determine the effect of incubation temperature on PiBV proliferation, PiBV infection cultures were performed at different incubation temperatures on each tissue fragment. For oysters reared at 16–18 °C (N = 8), six tissue fragments were taken from the marginal zone of the left mantle. For oysters reared at 23–25 °C (N = 12), eight tissue fragments were obtained from the marginal zones of the left and right mantles. After rinsing the mantles with sterile seawater, tissue fragments measuring approximately 3 mm wide and 5 mm long were excised and placed into separate wells of a 24-well plate. Each well was then filled with 1 mL of culture seawater and 10 μL of the infection source. The fragments from the oysters that were reared at 16–18 °C were then incubated at 20, 22, 25, 27, 30, and 32 °C, while those from the oysters reared at 23–25 °C were incubated at 15, 17.5, 20, 22.5, 25, 27.5, 30, and 32.5 °C for 7 days. Following the incubation period, the amount of PiBV genome in the cultured seawater was quantified by RT-qPCR.

Additionally, to assess the impact of cultivation temperature and incubation period on the viability of mantle epithelial cells, the marginal zones of the mantles from 8 oysters reared at 23–25 °C were cultured without PiBV infection. Tissue fragments were placed into the wells of a 24-well plate, with each well containing 1 mL of culture seawater. The samples were then incubated at 15, 17.5, 20, 22.5, 25, 27.5, 30, and 32.5 °C for 14 days. Daily observations were made using an inverted microscope to check for the presence of ciliary movement on the surface of the mantle tissue.

### 2.7. Histological Analysis of Cultured Uninfected Mantle Tissue

Six oysters that had been reared at 23–25 °C were used. After the marginal zone of the left mantle was trimmed and rinsed with sterile seawater, tissue fragments measuring approximately 3 mm wide and 5 mm long were excised. One section was immediately preserved in Davidson’s fixative [19] after trimming the mantle. Each fragment was placed in a separate well of a 24-well plate. To each well, 1 mL of culture seawater was added, and the plates were incubated at 25 °C. Tissue fragments from each individual were fixed on days 2, 4, 7, and 10 of cultivation. Paraffin sections of 3 µm thickness were prepared using a standard method and stained with hematoxylin and eosin. Sections were observed under a light microscope (BX53; Olympus, Tokyo, Japan) and imaged with a CCD camera (DP73; Olympus, Japan).

### 2.8. RT-qPCR-Based TCID_50_ Measurement

The TCID_50_ was measured using tissue fragments from six oysters reared at 16–18 °C and six oysters reared at 23–25 °C. Thirty tissue fragments measuring approximately 3 mm wide and 3 mm long (i.e., 6 individuals × 30 fragments = 180 fragments) were excised from the marginal zone of the mantle from each oyster. The samples were pooled and then arranged in six sets of 6 rows and 5 columns in separate wells of 96-well plates. Each well was filled with 100 μL of culture seawater. The infection source was serially diluted 10-fold in culture seawater to give well concentrations ranging from 10^−1^ to 10^−4^. Each sample dilution was performed in sextuplicate by adding 10 μL of diluted infectious source to six rows of wells containing the mantle tissue. For the negative controls, 10 μL of culture seawater without PiBV was added to a separate column of wells. Additionally, duplicate rows of wells containing only the serial dilutions without mantle tissue were included as an additional negative control. The plates were incubated at 25 °C for 7 days, after which the presence of PiBV genomic RNA in the culture seawater was detected by RT-qPCR. The TCID_50_ values were calculated using the Spearman–Kärber method [20,21].

### 2.9. Inactivation by Ultraviolet (UV) Light

A deep ultraviolet light-emitting diode (DUV-LED) device (Nikkiso Co., Ltd., Tokyo, Japan), producing a narrow-band wavelength of 280 ± 5 nm, was used as the light source. A total of 285 μL of infectious source and a stirrer bar (φ2 × 7 mm) were placed in a 0.95 cm^2^ dish to achieve a depth of 3 mm. The mixture was stirred at 300 rpm, and continuous irradiation was applied from a distance of 30 mm for varying durations (i.e., 0, 5, 10, 15, 20, 25, 30, 35, 40, 45, 50, and 55 s) at room temperature (25 °C). The light intensity at the surface of the viral suspension was 12.3 mW/cm^2^. Post-irradiation, the UV-treated PiBV solution was preserved on ice for subsequent TCID_50_ measurement. To determine the TCID_50_, 96 tissue fragments measuring approximately 3 × 3 mm in size were excised from the marginal zones of the mantles of four individual pearl oysters. The samples were pooled and randomly placed in 96-well plates.

Ten-fold serial dilutions from the original LED-UV irradiated samples (up to a 10^−4^ dilution) were prepared and inoculated onto the tissue fragments in triplicate, followed by incubation at 25 °C for 7 days. The presence of PiBV in the supernatant was detected by quantitative PCR, and the TCID_50_ was determined as described in Section 2.7. The inactivation efficiency of the UV irradiation was estimated using the formula (N0/Nt) × 100, where N0 and Nt are the TCID_50_ values at time zero and time *t* post-UV-LED irradiation, respectively.

### 2.10. Statistical Analysis

All statistical analyses were performed using the R software package (v4.4.1) [22]. Comparisons between the two groups were estimated using the Mann–Whitney *U* test using the ‘coin’ package (v1.4-3) [23]. Multiple comparisons were performed using the Steel–Dwass test implemented in the NSM3 package (v1.18) [24]. A value of *p* < 0.05 was considered statistically significant.

## 3. Results

### 3.1. Mantle Regions Suitable for PiBV Ex Vivo Propagation

In the marginal zone, PiBV replication was observed to exceed 1 × 10^4^ copies/μL by day 9 post-infection in all triplicate tissue fragments from all four individuals (Figure 2 top panel). In the pallial zone, the PiBV levels reached over 1 × 10^4^ copies/μL in all triplicate tissue fragments from two of the four oysters. Conversely, PiBV propagation was minimal in the central zone, except for one tissue sample from one oyster (Pearl Oyster 2). Throughout the cultivation periods, the PiBV levels in the central zone were significantly lower than those in the marginal and pallial zones (Figure 2, bottom panel). In the marginal and pallial zones, compared to day 2, the PiBV levels were significantly higher from day 4 onward. In the central zone, compared to days 2 and 4, the PiBV levels increased significantly from day 6 post-infection.

### 3.2. Reproducibility of PiBV Propagation Using the Marginal Zone of the Mantle

Among the four oysters tested, three oysters (Oysters 1, 3, and 4) showed PiBV copy numbers that varied within a 100-fold range across the 12 tissue fragments used for analysis (Figure 3). However, Oyster 2 exhibited a markedly wider variation of approximately 1000-fold in PiBV copy numbers in the culture medium. For Oyster 4, the mean PiBV copy numbers were significantly lower than those of the other oysters.

### 3.3. Effect of Cultivation Temperature on PiBV Propagation

Independent of the rearing water temperature, the highest PiBV copy numbers were observed at an incubation temperature of 25 °C (Figure 4). PiBV propagation was notably minimal at the temperatures of 15 °C and 32.5 °C.

When the mantle tissue fragments from eight oysters were cultured without PiBV infection, the ciliary movement on the surface of the mantle tissue fragments ceased by day 7 at 32.5 °C, by days 8–9 at 30 °C, by days 9–10 at 27.5 °C, by days 10–12 at 25 °C, by days 10–13 at 22.5 °C, and after day 13 at 20 °C. At 15 °C and 17.5 °C, ciliary movement was maintained in the mantle tissue fragments from all oysters, even at day 14.

### 3.4. Histological Analysis of Cultured Uninfected Mantle Tissue

In the six pearl oysters used in the experiment, no significant individual differences were observed in the results, and no time-dependent tissue changes were noted up to day 4 of the culture (Figure 5). On day 4, the six oysters showed no detachment of the epithelial cells, tissue atrophy or degeneration, or significant inflammation. The tissue structure remained generally normal, similar to that observed on days 0 and 2. On day 7, localized detachment of the epithelial cells, accompanied by cell death and tissue degeneration, was observed in the three individuals, with moderate pathological changes. In contrast, almost no tissue changes were observed in the other three oysters on day 7. However, by day 10, all six oysters exhibited mantle atrophy, and the three folds (inner, middle, and outer folds) at the edge of the mantle were not distinguishable. Additionally, most of the epithelium showed necrosis or detachment, and tissue degeneration was observed in the connective tissue. While ciliary structures of the epithelium were still present, inflammatory signs, including the accumulation of numerous hemocytes beneath the epithelium, were occasionally observed. Notably, no significant cell death or marked tissue dissolution was observed in the neural tissue, which is presumed to be relatively prone to degeneration, throughout the 10-day culture period.

### 3.5. RT-qPCR-Based TCID_50_ Measurement

The TCID_50_ values obtained from the oysters reared at 16–18 °C were significantly lower compared to those from the oysters reared at 23–25 °C (Figure 6). Specifically, the TCID_50_ values for the six oysters reared at 16–18 °C ranged from 1.5 × 10^3^ to 3.2 × 10^3^. In contrast, the values for the six oysters reared at 23–25 °C ranged from 2.2 × 10^3^ to 1.0 × 10^4^. The variation in the TCID_50_ values among individuals under the same conditions was within a 10-fold range.

### 3.6. Inactivation by UV Light

The TCID_50_ of PiBV demonstrated a linear decrease in response to increasing doses of UV irradiation, with a correlation coefficient of −0.82 (Figure 7). PiBV was 99% inactivated at a UV dose of 5530 J/m^2^. Furthermore, no PiBV propagation was detected following exposure to UV doses of 6150 J/m^2^ or higher.

## 4. Discussion

The propagation of PiBV was successfully conducted ex vivo using mantle tissue fragment cultures. This method facilitated the determination of optimal growth temperatures, TCID_50_ values, and UV sensitivity of the virus. To the best of our knowledge, apart from studies on *Aquabirnavirus*, which can be isolated from bivalves and propagated in fish cells [25,26,27], no other studies have assessed these parameters ex vivo for viruses that infect mollusks. While several studies have investigated the pathogenicity of oyster herpesvirus across different water temperatures [28,29,30], studies experimentally examining the optimal growth temperatures for viruses affecting mollusks appear to be lacking. Although PiBV grown in culture fragments has not been observed via electron microscopy, the increase in genome copy numbers provides clear evidence of viral proliferation.

In cases where cell lines are not available for mollusks, infection experiments using live organisms have traditionally been conducted to investigate viral characteristics. Such approaches require considerable labor, rearing space, and time. The ex vivo culture method established in this study enables various assays to be conducted with the same level of effort and scale as those using cultured cells. Additionally, because multiple tissue fragments can be prepared from a single oyster, it becomes feasible to compare different experimental conditions within the same individual. This approach permits the assessment of multiple test conditions within the same group of individuals, markedly reducing the influence of individual variability on the results. Because the outer mantle epithelial cells, where PiBV replicate, are culturable as primary culture cells and proliferate in vitro [17], experiments on PiBV using the cultured epithelial cells may be feasible. However, culturing mantle epithelial cells is a complex process, requiring several days for the cells to fully proliferate. The method using tissue explants established in this study is advantageous in that experiments can be initiated immediately by simply excising the mantle from the oyster. When using the marginal zone of the mantle at 25 °C, the viral load did not significantly change between days 4 and 9 post-infection (Figure 2 and Figure 4). In this study, most of the test parameters were evaluated based on the viral load in the culture medium on day 7 post-infection.

In this study, tissue fragments were cultured using seawater supplemented only with antibiotics as the culture medium. In research aiming for the long-term cell culture of mollusks, cell culture media containing nutrients with adjusted osmotic pressure are often used [9,10,12]. In a previous study similar to ours, the mantle tissues of oysters were cultured using Leibovitz’s medium-based solutions, confirming the ex vivo proliferation of OsHV-1 [15]. On the other hand, for short-term cell culture, seawater without nutrients [11,31] or buffers composed of inorganic salts [17] are sometimes employed. In this study, we used nutrient-free seawater as the culture medium to prevent bacterial growth due to the nutrients in the medium, considering that the study did not aim to promote cell growth during culture and that the culture period was short. The tissue fragments cultured using this method maintained ciliary movement up to the ninth day of culture at 25 °C. Histological observations revealed that, as of day 7 of culture, abnormalities were observed in the tissues. However, tissue disintegration is unlikely to be a significant issue for measuring PiBV proliferation because PiBV proliferates rapidly. Infection trials have confirmed that PiBV reaches its maximum load within 2–3 days post-infection [2,4]. Similarly, in this study, experiments conducted at a culture temperature of 25 °C demonstrated that the viral load in the supernatant from the marginal zone culture reached its maximum by the fourth day post-infection (Figure 2). Therefore, tissue damage occurring after 7 days of culture would likely not interfere with the assessment of PiBV proliferation. The mantle of the *Pinctada* genus is histologically divided into the marginal, pallial, and central zones [32]. Our findings show that PiBV replication was most pronounced in the marginal zone and minimal in the central zone. The three regions of the mantle perform different functions related to shell formation, which may explain the observed variations in PiBV susceptibility [33,34]. Specifically, the epithelial cells of the marginal zone are involved in the formation of the prismatic layer, whereas those in the pallial zone are responsible for nacreous layer formation and are crucial for pearl production when nuclei are transplanted into a donor oyster. The central zone also plays a role in nacreous layer formation. The different roles of these mantle regions in shell formation can further be explained by the variations in the morphology of the outer mantle epithelial cells [33,35] and the expression of specific genes in these regions [33,35,36,37,38,39]. PiBV primarily infects the outer mantle epithelial cells [2]. The functional specialization of epithelial cells in different mantle regions may account for the observed differences in PiBV susceptibility across these regions. There is a “hot spot” for the proliferation of outer epithelial cells in the central zone of the mantle of *Pinctada fucata*, suggesting the existence of a region-specific proliferative ability that diminishes progressively away from this center [35]. Nonetheless, the active replication of PiBV in the marginal zone, where the proliferative activity of outer epithelial cells is considered to be lower, suggests that factors beyond merely the proliferative capacity of these epithelial cells might influence PiBV infection and replication dynamics.

While no significant differences were observed in PiBV propagation between the marginal and the pallial zones, the marginal zone demonstrated less variability in PiBV replication among different individuals (Figure 2). Consequently, employing the marginal zone for ex vivo experiments can yield more consistent and reproducible results. In this study, the marginal zone was used to determine the optimal growth temperatures, measure the TCID_50_ values, and assess UV sensitivity.

PiBV propagation peaked at 25 °C, regardless of the rearing temperature of the oysters used in the experiments. Infection experiments conducted at a rearing water temperature of 18 °C showed minimal onset of disease, whereas mantle atrophy and mortality were evident at 23 °C and 28 °C [40]. The replication characteristics of PiBV observed in this study are consistent with the results of the infection experiments. In most of the pearl farming areas in Japan, there are periods when seawater temperatures exceed 30 °C. Although PiBV replication was not prominent at 30 °C in this study, could infection during high-temperature periods lead to mortality in pearl oysters? To clarify the status of the disease during these high-water temperature periods in areas where pearl culture is practiced, it is necessary to investigate the infection dynamics of PiBV and the onset of the disease during high-temperature periods. This could involve more focused infection experiments or epidemiological surveys in farming areas.

In the PiBV-uninfected mantle tissue fragments, ciliary movement ceased by day 7 when incubated at 32.5 °C, whereas at lower temperatures, ciliary movement persisted for at least seven days. Ciliary motility serves as an indicator of epithelial cell viability. The minimal PiBV replication observed at extremely low or high temperatures may not necessarily be due to cell death induced by suboptimal incubation temperatures, but rather due to a reduction in viral replication efficiency. Pearl oysters exhibit the fastest growth between 20 and 26 °C [41], and their filtration rate is highest between 25 and 28 °C, dropping significantly below 13 °C and above 31 °C [42]. This temperature profile closely follows the relationship between PiBV replication and different incubation temperatures. Since viral replication is highly dependent on the metabolic activity of host cells, the physiological activity of pearl oyster cells likely determines PiBV replication levels.

Relatively stable minimum and maximum TCID_50_ values, with minimal and maximal values falling within a five-fold range, were consistently observed in tissue fragments from six individuals maintained under identical rearing conditions. While the rearing water temperature did influence the TCID_50_ values of the pearl oysters, the standardized measurement conditions enabled reliable comparisons of virus solutions subjected to different treatments. In this study, TCID_50_ values were employed to assess the UV sensitivity of PiBV, which was completely inactivated at a UV dose of 6150 J/m^2^.

The UV inactivation thresholds of various viruses have been measured in previous studies, with members of the families *Birnaviridae* (such as *Aquabirnavirus*) and *Nodaviridae* exhibiting high resistance to UV light [43]. For example, Kitamura et al. [44] reported that an Aquabirnavirus isolated from pearl oysters was not inactivated by sunlight. Additionally, to achieve 99.9% inactivation of infectious pancreatic necrosis virus (IPNV), which belongs to the genus *Aquabirnavirus*, a UV dose of 1188 ± 57 J/m^2^ was required [45]. The UV resistance of PiBV is comparable to that of IPNV, suggesting that PiBV, like the genus *Aquabirnavirus*, appears to possess a high level of UV resistance.

Even under identical experimental conditions, substantial individual variation in PiBV copy numbers was observed. The ex vivo culture method might be advantageous for identifying individuals with enhanced disease resistance. Nonetheless, the pronounced individual differences could influence the outcomes of the experiments. Figure 3, which shows the PiBV copy numbers from 12 marginal zone tissue fragments from the left and right mantles of one individual, shows that two individuals exhibited significantly high PiBV copy numbers, one individual exhibited low copy numbers, and another displayed considerable variation among tissue fragments. Unlike cell lines, which consist of homogeneous cell types and enable highly reproducible tests by controlling culture conditions, tissue fragment cultures do not offer such reproducibility. Although the histological structure of the mantle’s outer edge in the genus *Pinctada* shows minimal variation along the edge [46], there appeared to be biological differences among individuals or among different regions of the same individual. To mitigate the impact of large differences among individuals, experiments using ex vivo tissue fragment culture methods should involve testing with multiple individuals. However, by accounting for inter-individual or site-specific differences and employing multiple tissue fragment samples, this tissue fragment culture method can be used to perform ex vivo experiments and could potentially accelerate PiBV research.

## Figures and Tables

**Figure 1 pathogens-14-00076-f001:**
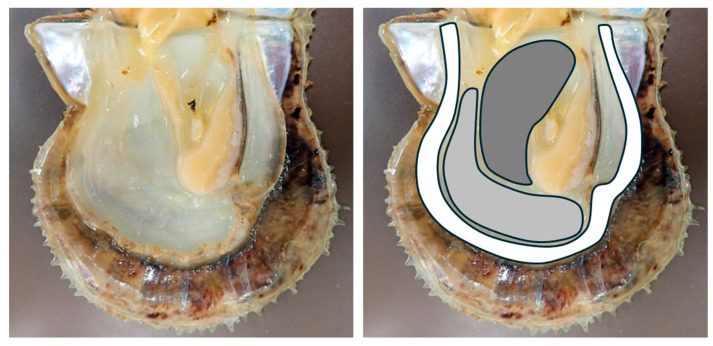
The image on the left shows the shell of a pearl oyster with the visceral mass removed. The image on the right shows the mantle region excised for this study. The regions are color-coded as follows: White: marginal zone, gray: pallial zone, dark gray: central zone.

**Figure 2 pathogens-14-00076-f002:**
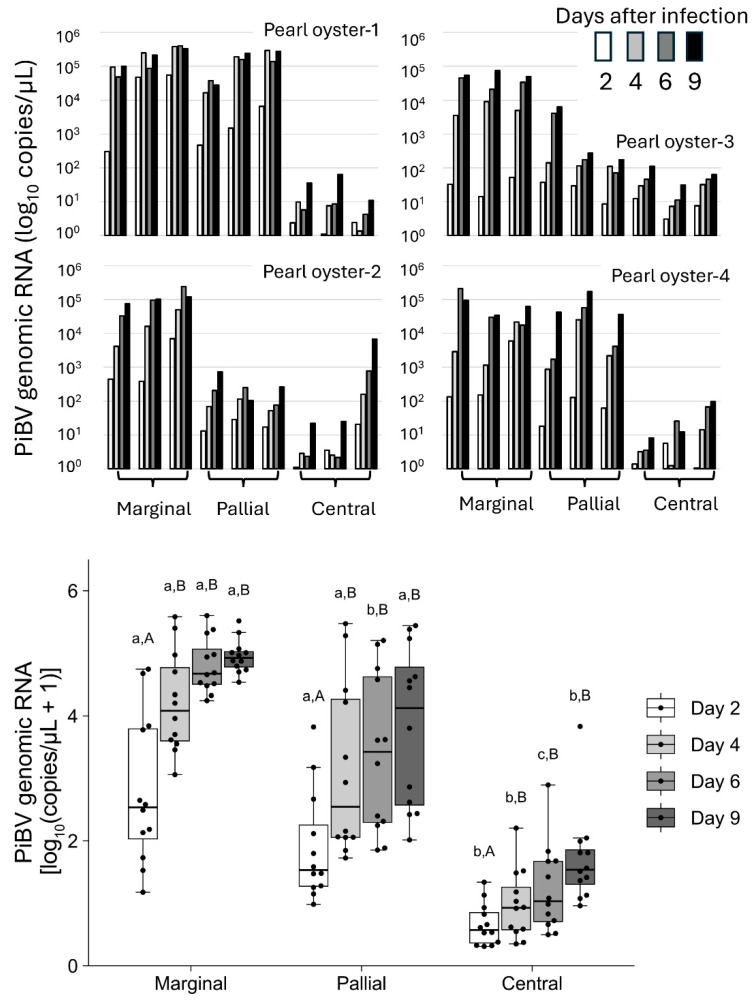
**Top panel:** The PiBV copy number in individual pearl oysters (N = 4). Ex vivo cultivation was performed using three tissue fragments from the marginal, pallial, and central zones of the mantle from each oyster. The copy number of PiBV in the culture seawater was quantified on days 2, 4, 6, and 9 post-infection. The white bars represent the PiBV copy number on day 2, the light gray bars represent day 4, the gray bars represent day 6, and the black bars represent day 9. **Bottom panel:** Box plots showing the PiBV copy numbers from the marginal, pallial, and central zones of all four oysters using a total of 12 tissue fragments. The black dots indicate the actual PiBV levels in each tissue fragment. Significant differences among the three tissue zones on the same sampling day are indicated by three lowercase letters, while significant differences within the same tissue zone across the four sampling days are denoted by different uppercase letters (Steel–Dwass test, *p* < 0.05). The color coding of the bars is consistent with that described for the top panel: the white bars represent the PiBV copy number on day 2, the light gray bars represent day 4, the gray bars represent day 6, and the dark gray bars represent day 9. The boxes in the box plots indicate the interquartile range (IQR) with the median, and the whiskers extend to the most extreme data points within 1.5 times the IQR from the lower or upper quartile.

**Figure 3 pathogens-14-00076-f003:**
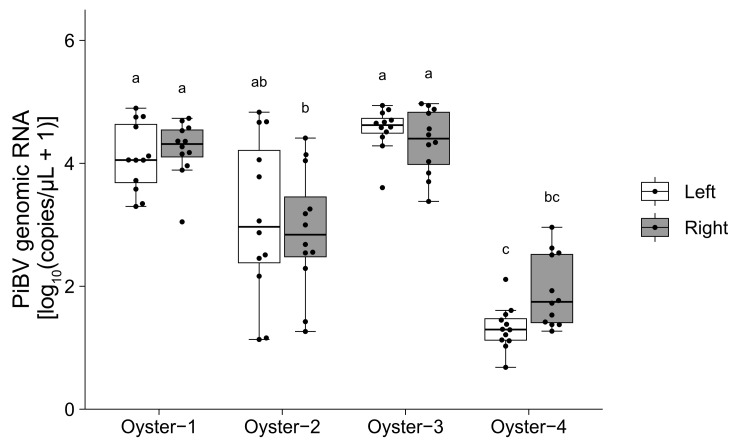
Ex vivo cultivation was performed using 12 mantle fragments from both the left and right sides of four oysters. The PiBV copy number in the culture medium was measured on day 7 post-infection. The open box represents the left mantle, while the gray box represents the right mantle. The black dots indicate the actual PiBV copy numbers in each tissue culture. Different letters denote significant differences among the different groups (Steel–Dwass test, *p* < 0.05). The boxes in the box plots indicate the interquartile range (IQR) with the median, and the whiskers extend to the most extreme data points within 1.5 times the IQR from the lower or upper quartile.

**Figure 4 pathogens-14-00076-f004:**
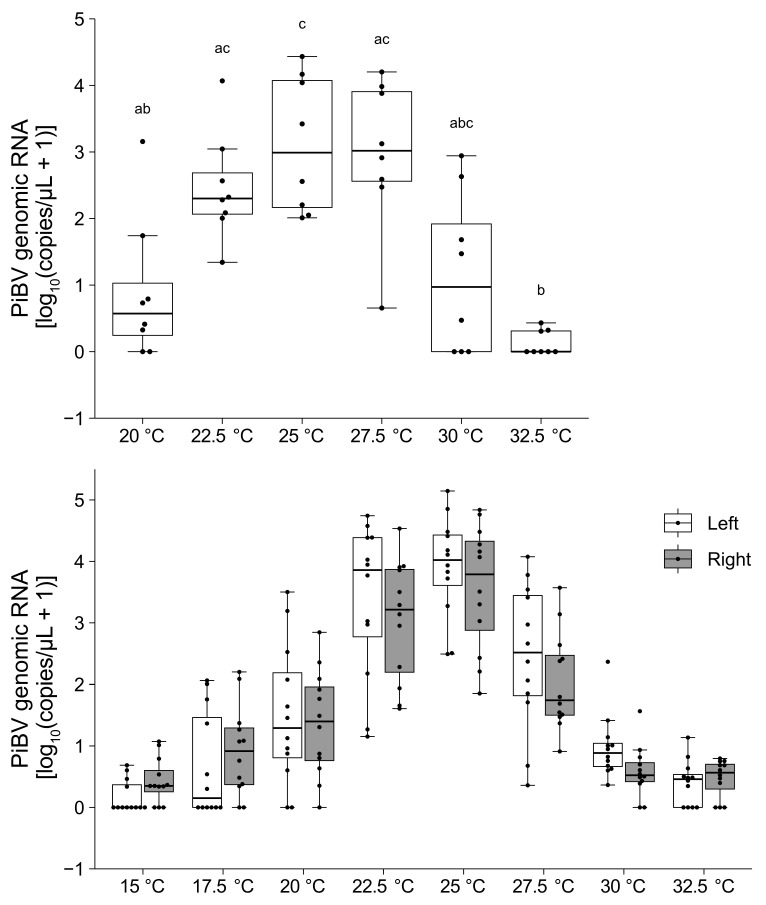
The PiBV copy number was measured on day 7 post-infection after ex vivo cultivation at different temperatures. **Top panel:** The results using pearl oysters reared at 16–18 °C (N = 8). Different combinations of letters indicate significant differences, as estimated by the Steel–Dwass test (*p* < 0.05). **Bottom panel:** The results from pearl oysters reared at 23–25 °C (N = 12). White and gray represent the left and right mantle, respectively. Detailed statistical test results are provided in Appendix A. The boxes in the box plots indicate the interquartile range (IQR) with the median, and the whiskers extend to the most extreme data points within 1.5 times the IQR from the lower or upper quartile.

**Figure 5 pathogens-14-00076-f005:**
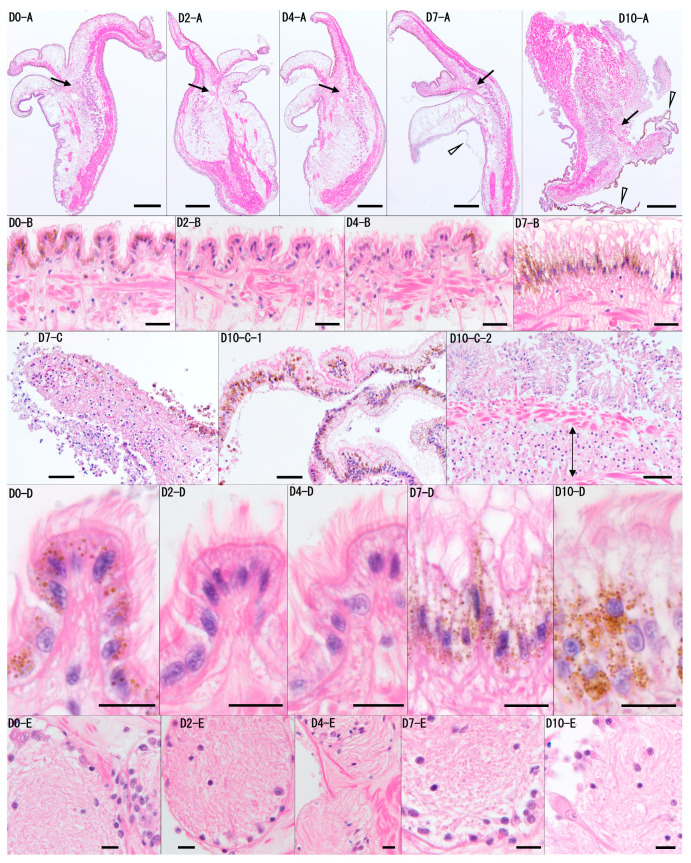
Representative histological images of the mantle tissue from Pearl Oyster 6 on days 0, 2, 4, 7, and 10 (D0, D2, D4, D7, D10) are shown. Panel (**A**) presents low-magnification cross-sectional images of the mantle, while panels (**B**–**D**) are higher magnifications of the outer layer of the mantle shown in (**A**). Panel (**D**) is a higher magnification of panel (**B**), and panel (**E**) shows a high-magnification image of the neural tissue (arrow) from panel (**A**). No time-dependent tissue changes were observed on day 0, day 2, or day 4. On day 7, localized detachment of the epithelial cells was noted (D7-C). The arrowhead in D7-A indicates the detached epidermis, which corresponds to nearby detached tissue. On D10, the mantle exhibited atrophy, and the three folds (inner, middle, and outer folds) at the edge of the mantle were not distinguishable (D10-A). Additionally, on day 10, the epidermis showed widespread detachment and folding (arrowheads in D10-A, D10-C-1), and an accumulation of hemocytes beneath the epidermis was observed (arrowed area in D10-C-2). However, throughout the 10-day period, the ciliary structures on the outermost layer of the epithelial cells were still recognizable (D0, D2, D4, D7, D10-B/D). In the neural tissue, no significant cell death or marked tissue dissolution was observed (D0, D2, D4, D7, D10-D). Scale bars: (**A**), 400 μm; (**B**), 20 μm; (**C**), 40 μm; (**D**,**E**), 10 μm. Staining: H&E.

**Figure 6 pathogens-14-00076-f006:**
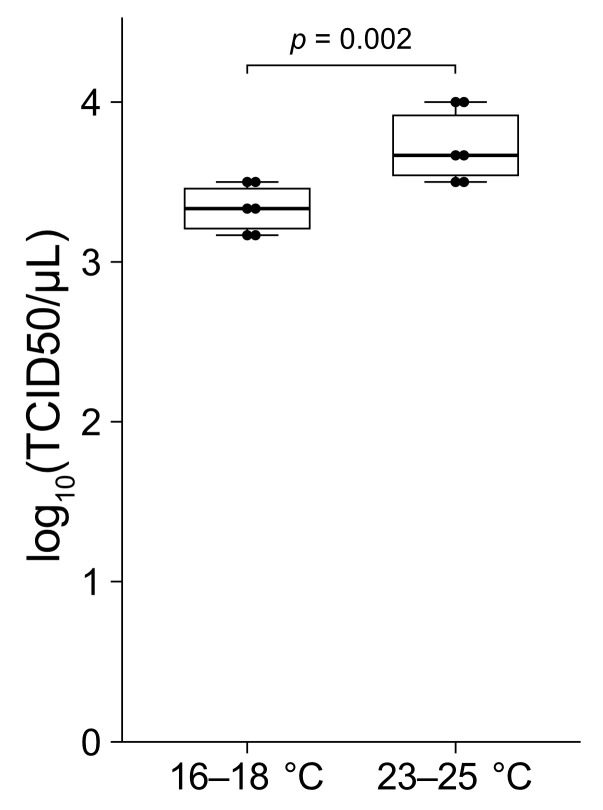
The mantle tissue fragments from the marginal zones of six pearl oysters reared at 16–18 °C and six oysters reared at 23–25 °C were used in this study. The tissue samples were infected with serial dilutions of PiBV and incubated at 25 °C. PiBV infection was assayed at day 7 post-infection to determine the TCID_50_. The two groups were compared using the Mann–Whitney *U* test. The boxes in the box plots indicate the interquartile range (IQR) with the median, and the whiskers extend to the most extreme data points within 1.5 times the IQR from the lower or upper quartile.

**Figure 7 pathogens-14-00076-f007:**
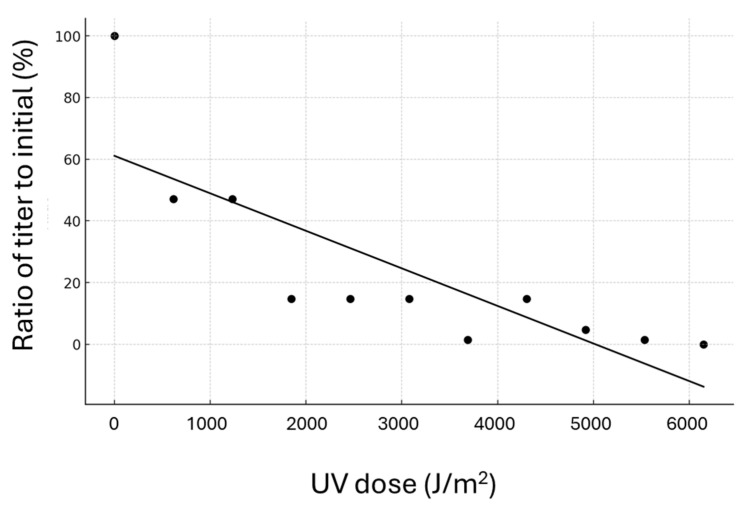
Each point on the graph shows the ratio of the titer following UV radiation treatment to the initial titer. Tissue fragments excised from the marginal zone of the mantle of four individual pearl oysters were pooled and randomly distributed into 96-well plates to determine the TCID50.

## Data Availability

Please contact the corresponding author for access to raw data.

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
