# Peer review of "Ex Vivo Propagation of Pinctada Birnavirus Using Mantle Tissue Fragment Culture: Application for Measuring Replication at Different Temperatures, TCID50 Assay, and UV Sensitivity"

_pathogens, 2025, doi:10.3390/pathogens14010076_

Round 1
Reviewer 1 Report
Comments and Suggestions for Authors
The manuscript describes a novel method for culturing mussel viruses, which if accurately described, represents an incredible find. However, I have questions about the methods.
The manuscript was well written and only edit I found would be to change it to "CHSE-214" on line 37.
Honestly the methods seem too good to be true and perhaps something is missing. The authors state that they kept mussel tissues alive and without contamination using only sterile seawater with antibiotics (without any culture medium/energy source) for 9 days. In my experience attempting to culture mussel cells, this seems virtually impossible. However, they do have proof that the virus was cultured in the tissue explants as the qPCR values increased. So i am not sure what to think about the methods. I would like the authors to add more details regarding whether or not contamination occurred and add any references to support that their methods are possible (i.e.: has anyone kept mussel cells alive using only sterile sea water). If these methods are accurate, this study represents an incredible find.
Author Response
We appreciate the reviewer’s acknowledging the scientific importance of this manuscript and helpful comments. The manuscript has been revised according to the comments, as follows.
In this study, high concentrations of antibiotics were used to the culture seawater to inhibit bacterial growth. However, this did not result in a completely sterile culture. Additionally, as explained in lines 275–280, higher culture temperatures appear to cause cells to die in a shorter period of time. Even at 25°C, the culture temperature used in many experiments in this study, ciliary movement of epithelial cells ceased within 10–12 days of culture. Thus, the conditions are far from what is typically considered optimal for cell culture.
Nevertheless, since PiBV proliferates rapidly, it is feasible to conduct PiBV experiments as long as the cells remain viable for approximately 10 days.
As a response to your comments, and following the suggestions of Reviewer 2, we conducted histological observations of mantle tissue fragments cultured in seawater (Line 177-186, 291-322, 379-396). Additionally, we cited studies that cultured homocyte or tissues in seawater similarly to this study (Line 383-385).
Also corrected CHSE214 to CHSE-214 (Line 36).
Reviewer 2 Report
Comments and Suggestions for Authors
The manuscript by Matsuyama et al. presents a study on the ex vivo propagation of Pinctada birnavirus (PiBV) using mantle tissue fragment culture. The research is significant as it attempts to overcome the limitations in studying Mollusk viruses due to the lack of suitable cell lines for their propagation. The authors investigated the virus's proliferation characteristics at different temperatures, established a TCID50 assay, and examined its UV sensitivity.
Major Comments
1. The author should review the research progress of invertebrate cell lines and tissue culture methods in the discussion section by integrating the references 9 - 15, 17, 24 - 26, and any other related refs. It is necessary to detail which invertebrate cells or tissues have been successfully cultured, along with their specific culture conditions. Specially, comparisons should be made between these established methods and the mantle tissue culture method in this study. Additionally, an explanation should be provided as to why only seawater was used for the mantle tissue culture in this research. This will help readers better understand the novelty and significance of the current study in the context of existing research.
2. To enhance the clarity and reliability of the tissue state assessment during the mantle tissue culture process, it is recommended that the author provide microscopic photos at various time points. These photos are crucial for validating the experimental results and the physiological state of the cultured mantle tissue.
3. It is not clear whether TEM observation was conducted in this study. Visualizing the viral particles proliferating in cells under the electron microscope would provide direct and compelling evidence to support the claim that the virus replicates intracellularly rather than simply attaching to the outside of the tissue.
4. The unit of the qPCR detection results in this study, which is currently "copies/µL", is inappropriate. A more suitable unit would be "copies/ng host DNA" or "copies/mg tissues". This adjustment would provide a more accurate and meaningful quantification of the virus in relation to the host tissue, allowing for better comparison and interpretation of the data without the study.
5. In the TCID50 assay, the author used 3×3 mm tissue blocks, while in the previous culture experiments, 3×5 mm tissue blocks were employed. The reason for this difference in tissue block size should be clearly explained.
6. Although the author has described the repetition numbers in the materials and methods section, it is essential that all figures in the paper explicitly indicate the repetition numbers of the oyster individuals and tissue blocks used in each experiment. This will provide transparency and clarity, allowing readers to easily assess the reliability and statistical significance of the data presented.
Author Response
We appreciate the reviewer’s acknowledging the scientific importance of this manuscript and helpful comments. The manuscript has been revised according to the comments, as follows.
Major Comments
- The author should review the research progress of invertebrate cell lines and tissue culture methods in the discussion section by integrating the references 9 - 15, 17, 24 - 26, and any other related refs. It is necessary to detail which invertebrate cells or tissues have been successfully cultured, along with their specific culture conditions. Specially, comparisons should be made between these established methods and the mantle tissue culture method in this study. Additionally, an explanation should be provided as to why only seawater was used for the mantle tissue culture in this research. This will help readers better understand the novelty and significance of the current study in the context of existing research.
Discussion of shellfish tissue or cell culture is included in line 379-396 of the revised manuscript.
- To enhance the clarity and reliability of the tissue state assessment during the mantle tissue culture process, it is recommended that the author provide microscopic photos at various time points. These photos are crucial for validating the experimental results and the physiological state of the cultured mantle tissue.
Histological observations were conducted on the cultured tissue fragments, accompanied by explanations and figures (lines 177–186, 291–322, 379–396).
- It is not clear whether TEM observation was conducted in this study. Visualizing the viral particles proliferating in cells under the electron microscope would provide direct and compelling evidence to support the claim that the virus replicates intracellularly rather than simply attaching to the outside of the tissue.
Although TEM observations have not yet been performed, there is no doubt that the virus is increasing as the PiBV genome increases. We have added this explanation to Line 359-361.
- The unit of the qPCR detection results in this study, which is currently "copies/µL", is inappropriate. A more suitable unit would be "copies/ng host DNA" or "copies/mg tissues". This adjustment would provide a more accurate and meaningful quantification of the virus in relation to the host tissue, allowing for better comparison and interpretation of the data without the study.
Since the amount of virus in the culture supernatant was measured in this study, it is shown as the amount per µL of supernatant measured. Line99-101 has been modified to show this as follows.
“The concentration of PiBV genomic RNA in each culture supernatant was calculated using the following formula and expressed as the number of PiBV copies per μL of culture seawater: (Number of virus copies detected × Total amount of RNA extracted from the sample) / Amount of RNA used in the reaction mixture.”
- In the TCID50 assay, the author used 3×3 mm tissue blocks, while in the previous culture experiments, 3×5 mm tissue blocks were employed. The reason for this difference in tissue block size should be clearly explained.
The TCID50 assay was conducted using 96-well plates, where 30 tissue fragments were excised from a single individual. In contrast, other culture experiments used 24-well plates, with 3 to 12 tissue fragments prepared from a single individual for testing. The reason for using smaller tissue fragments in the TCID50 assay was to match the size of the wells and to obtain a larger number of fragments from a single individual. The variation in tissue fragment sizes was not critical to this study, and therefore, it was not mentioned in the main text.
- Although the author has described the repetition numbers in the materials and methods section, it is essential that all figures in the paper explicitly indicate the repetition numbers of the oyster individuals and tissue blocks used in each experiment. This will provide transparency and clarity, allowing readers to easily assess the reliability and statistical significance of the data presented.
We have added the number of individuals and tissue fragments to the descriptions of Figure 4 and Figure 6, as these details were not previously included.
Round 2
Reviewer 1 Report
Comments and Suggestions for Authors
I believe they have addressed my concerns and the article is suitable for publication.